# Camel Milk-Derived Extracellular Vesicles as a Functional Food Component Ameliorate Hypobaric Hypoxia-Induced Colonic Injury Through Microbiota–Metabolite Crosstalk

**DOI:** 10.3390/nu17152431

**Published:** 2025-07-25

**Authors:** Hui Yang, Demtu Er, Yu-Huan Wang, Bin-Tao Zhai, Rili Ge

**Affiliations:** 1College of Medical, Qinghai University, Xining 810016, China; yanghui@qhu.edu.cn (H.Y.);; 2College of Veterinary Medicine, Inner Mongolia Agricultural University, Hohhot 010018, China; eedmt@imau.edu.cn; 3Lanzhou Institute of Husbandry and Pharmaceutical Sciences, Chinese Academy of Agricultural Sciences, Lanzhou 730046, China; zhaibintao@163.com

**Keywords:** high-altitude hypoxia, colonic injury, camel milk extracellular vesicles (CM-EVs), gut microbiota, metabolome

## Abstract

**Background/Objectives:** This study investigates the therapeutic potential of camel milk-derived extracellular vesicles (CM-EVs) for treating colonic damage caused by high-altitude hypoxia, supporting the WHO’s “Food as Medicine” initiative. **Methods:** Using a 5500 m mouse model, researchers induced colonic injury and treated it with oral CM-EVs for 15 days, comparing results to whole camel milk. **Results:** CM-EVs outperformed whole milk, significantly improving colon health by restoring barrier integrity and reducing disease activity index (DAI) (*p* < 0.01). They boosted beneficial bacteria like *Lactobacillus* and *Bifidobacterium* and decreased *Enterobacteriaceae* (*p* < 0.01). Metabolic analysis showed restored bile acid balance and amino acid modulation via the FXR/NF-κB pathway, reducing TLR4/MyD88-mediated inflammation and oxidative stress (*p* < 0.01). Fecal microbiota transplantation in the CM-EVs group notably decreased DAI and increased colon length (*p* < 0.05). **Conclusions:** CM-EVs repair mucosal damage, balance microbiota, and regulate metabolism to combat hypoxia-induced colonic damage, suggesting their potential as nutraceuticals and altitude-adaptive foods. This showcases nanotechnology’s role in enhancing traditional dietary benefits via precision nutrition.

## 1. Introduction

Acute mountain sickness (AMS), precipitated by rapid ascension to altitudes exceeding 2500 m, presents with gastrointestinal dysfunction and systemic complications that significantly hinder high-altitude acclimatization [1]. This condition poses a substantial challenge for global mountain expeditions and populations susceptible to hypoxia [2]. Although conventional pharmacological treatments are available, there is an increasing interest in food-derived therapeutics, which aligns with the World Health Organization’s “Food as Medicine” initiative, especially in the context of managing environmentally induced pathologies [3,4,5]. Hypobaric hypoxia intensifies colonic mucosal injury through oxidative stress and NF-κB-mediated inflammation, while simultaneously depleting endogenous antioxidants and microbial metabolites, such as butyrate, which are crucial for epithelial repair [6,7,8]. This duality underscores the gut as a strategic target for interventions based on functional foods.

Emerging evidence highlights extracellular vesicles (EVs) from medicinal foods as next-generation nutraceutical carriers [9]. Structurally, EVs are encapsulated by a lipid bilayer rich in cholesterol, sphingomyelin, and ceramides, and they transport cell-specific cargoes, including proteins such as conserved tetraspanins (CD63, CD81, and CD9), nucleic acids (mRNA, noncoding RNAs, and DNA), and bioactive lipids [10,11,12,13,14]. Functionally, EVs serve as versatile signaling mediators by transferring biomolecules to recipient cells, thereby influencing processes such as extracellular matrix remodeling, apoptosis, and immune responses [15,16,17]. The effects of EVs are context-dependent, determined by the surface receptor profiles of target cells and the compositional heterogeneity of the EVs themselves [18]. This heterogeneity originates from diverse cellular sources, such as dendritic and epithelial cells, while maintaining conserved protein markers that facilitate functional diversity in both physiological and pathological contexts [18]. EVs are notably pervasive in eukaryotic body fluids, including blood, urine, and saliva, and they play dual roles in disease progression and therapeutic development. They are involved in the pathogenesis of neurodegenerative diseases, metabolic disorders, and cancer [19,20,21], yet they also offer potential as diagnostic biomarkers and drug delivery systems due to their biocompatibility and target specificity [18,19,20,21].

Camel milk, a traditional ethnopharmacological resource in arid regions, harbors nanosized EVs (CM-EVs, 30–200 nm) enriched with evolutionarily conserved bioactive cargo (miRNAs, phosphatidylserine, and immunomodulatory proteins) [22]. Unlike synthetic nanoparticles, CM-EVs inherently embody the “medicinal-food homology” principle—their lipid–protein architecture synergizes with host physiology to modulate microbiota–metabolite crosstalk, offering unparalleled biocompatibility [23]. Notably, CM-EVs contain hypoxia-responsive RNAs, such as MIR-101 and MIR-19b-3p, which facilitate epithelial repair and reduce the production of pro-inflammatory cytokines [23,24,25]. These attributes suggest that CM-EVs may effectively mitigate altitude sickness-associated colonic injury through the integrated modulation of microbiota, metabolites, and immune responses, although further mechanistic studies are required to substantiate these effects.

This study elucidates the potential of CM-EVs as bioactive dietary components to ameliorate hypoxia-induced colonic dysfunction through targeted modulation of the microbiota, regulation of Short Chain Fatty Acids (SCFAs) metabolism, and suppression of inflammation. These findings position CM-EVs as novel functional food agents, providing a sustainable strategy to mitigate altitude-associated gastrointestinal compromise via diet-microbiota interactions. By integrating traditional ethnopharmacological knowledge of the “desert adaptation diet” with foodomics-driven nanotechnology, this research redefines CM-EVs as WHO-aligned functional food vectors, offering a transformative framework for developing altitude-adaptive nutraceuticals through agriculture-sourced, microbiota-directed precision nutrition.

## 2. Materials and Methods

### 2.1. Animal Ethics and Experimental Design

Male C57BL/6J mice (6–8 weeks old, SPF grade; SCXK 2024-0012) were obtained from Jiangsu Huachuang Biotechnology (Zhenjiang, China). After 7-day acclimatization in ventilated cages (Tecniplast GM500, Buguggiate, Italy) under controlled conditions (20 ± 1 °C, 50 ± 5% humidity, 12 h light/dark cycle), mice received ad libitum AIN-93G diet (Trophic Animal Feed, Nantong, China) and autoclaved water. Body weight and intake were monitored daily. The specific experimental design is shown in Figure 1. Protocols followed China’s Guidelines for Ethical Review of Animal Welfare (GB/T 35892-2018) [25] and were approved by Qinghai University’s IACUC (PJ202501-55).

### 2.2. Optimized Protocol for Camel Milk Extracellular Vesicle (CM-EVs) Isolation

During the lactation period in May 2024, fresh raw milk was procured from free-ranging Qaidam Bactrian Camel herds, specifically firstborn, in Ulan County, Haixi Mongol and Tibetan Autonomous Prefecture, Qinghai Province, at an altitude of 3200 m (GPS coordinates: 37.37° N, 97.37° E). The milk was aliquoted into RNase-free cryovials (Corning, Corning, NY, USA, Cat# 430659), rapidly frozen using dry ice, and subsequently stored at −80 °C in a Thermo Scientific Forma 900 Series freezer (Thermo, Waltham, MA, USA). The isolation of CM-EVs was conducted according to a protocol adapted from Théry et al. (Journal of Extracellular Vesicles, 2018) [26]. Specifically, 50 mL of raw milk was treated with 0.05% (*w*/*v*) rennet (Chr. Hansen CHY-MAX^®^ M 1000, Hørsholm, Denmark), incubated at 4 °C, and subjected to a series of centrifugation steps using a Beckman Coulter (Brea, CA, USA) Avanti J-26S XP centrifuge with a JA-10 rotor (5000× *g* for 10 min; 16,500× *g* for 60 min) and a Type 70 Ti rotor (100,000× *g* for 90 min). The purified CM-EVs were resuspended in phosphate-buffered saline (PBS; Gibco, Billings, MT, USA, 10010023), filtered through 0.22 μm polyvinylidene fluoride (PVDF) membranes (Millipore, Burlington, MA, USA, SLGV033RS), and quantified using a bicinchoninic acid (BCA) assay (Keygen Biotech, Nanjing, China, KGP902) with bovine serum albumin (BSA) standards (Sigma-Aldrich, Saint Louis, MO, USA, A7906).

### 2.3. Comprehensive Characterization of CM-EVs

The CM-EVs were subjected to negative staining with 2% uranyl acetate and subsequently imaged using a Hitachi HT-7800 transmission electron microscope (Hitachi, Tokyo, Japan), operating at an acceleration voltage of 80 kV. The particle size distribution and concentration were assessed using a ZetaView PMX-120 system (Particle Metrix, Inning am Ammersee, Germany) in scatter mode, with a laser wavelength of 488 nm, a sensitivity setting of 85, and a frame rate of 30 frames per second. Protein analysis was conducted via SDS-PAGE using a 12% gel, followed by Coomassie Blue staining (Solarbio, Beijing, China). Western blotting was performed to detect exosomal markers using specific antibodies: CD63 (Abcam, Dalian, China, ab134045, dilution 1:1000), CD81 (Abcam, Dalian, China, ab109201, dilution 1:1000), and TSG101 (Abcam, Dalian, China, ab125011, dilution 1:500).

### 2.4. Hypobaric Hypoxia-Induced AMS Model and Disease Activity Index (DAI) Evaluation

Male C57BL/6J mice (*n* = 6 per group) were randomly allocated to experimental conditions and subjected to a simulated altitude of 5500 m, characterized by a barometric pressure of 379 ± 2 mmHg and an oxygen concentration of 10.8 ± 0.3%, within a hypobaric chamber (Fenglei FL-HPC3000, Guizhou Fenglei Aviation Equipment Co., Ltd., Anshun, China). The experimental groups were as follows: (1) hypoxia control (hypoxia + 50 μL/kg/day PBS); (2) hypoxia with camel milk supplementation (50 μL/kg/day raw camel milk); and (3) hypoxia with camel milk-derived extracellular vesicles (CM-EVs) (0.5 mg/kg CM-EVs in 50 μL/kg/day PBS). The respective treatments were administered for 15 consecutive days prior to exposure to hypoxia. Evaluate and calculate the DAI score based on literature [16] to comprehensively evaluate the severity of gut ulcer thoroughly. DAI = (weight loss + fecal characteristics + blood stool)/3. Calculate the weight, bowel movements, and bleeding status of mice (Urine fecal occult blood test kit, Regen Biotechnology Co., Ltd., Beijing, China).

### 2.5. Gut Microbiota Analysis via 16S rRNA Sequencing

Fecal DNA was extracted utilizing the OMEGA DNA Kit (D5625-01, Omega Bio-Tek, Norcross, GA, USA) in accordance with the manufacturer’s protocol. The V3-V4 hypervariable region of the bacterial 16S rRNA gene was amplified using primers 338F (5′-ACTCCTACGGGAGGCAGCAG-3′) and 806R (5′-GGACTACHVGGGTWTCTAAT-3′), synthesized by Tsingke Biotechnology Co., Ltd., Beijing, China. Library preparation was conducted with the TruSeq Nano DNA LT Kit (Illumina, San Diego, CA, USA, Cat# 20015964), followed by sequencing on an Illumina NovaSeq 6000 platform in a 2 × 250 bp paired-end mode using the NovaSeq 6000 S4 Reagent Kit v1.5 (300 cycles). Raw sequencing data were processed using QIIME2 (version 2021.4; https://qiime2.org) with the DADA2 plugin (version 2021.4.0) for denoising and Amplicon Sequence Variant (ASV) clustering at 100% similarity. Taxonomic assignment was conducted against the SILVA v138 reference database (https://www.arb-silva.de) with a confidence threshold of 99%. Alpha diversity indices, including Shannon and Chao1 and beta diversity, assessed using Bray–Curtis distance and Principal Coordinate Analysis (PCoA), were calculated via the QIIME2 core metrics pipeline and visualized using the R packages vegan (version 2.6-4) and ggplot2 (version 3.4.0).

### 2.6. Metabolomic Profiling and Pathway Analysis

Fecal metabolites were extracted using 80% methanol (LC-MS grade, Merck Millipore, Darmstadt, Germany, 106035) through vortexing for 10 min (Scientific Industries Vortex-Genie 2, Bohemia, NY, USA), followed by ultrasonication for 30 min at 4 °C and 40 kHz (Sonics VCX-750, Newtown, CT, USA). Liquid chromatography–mass spectrometry (LC-MS) analysis was conducted utilizing a Shimadzu LC-30AD system (Shimadzu, Tokyo, Japan) in conjunction with a SCIEX TripleTOF 5600 + mass spectrometer (SCIEX, Framingham, MA, USA, Serial No. HU-2023-ABX) equipped with an electrospray ionization (ESI) source. Chromatographic separation was achieved using a Waters HSS T3 column (2.1 × 100 mm, 1.8 μm; Waters, Milford, MA, USA, 186007494), employing mobile phases consisting of (A) 0.1% formic acid (Sigma-Aldrich, USA, 56302) and (B) acetonitrile (Honeywell, Charlotte, NC, USA, 34998). Mass spectrometry parameters included both positive and negative ion modes, a spray voltage of ±5.5 kV, a gas temperature of 500 °C, and a scan range of *m*/*z* 50–1000 with a resolution of 35,000 FWHM. The raw data were processed using MS-DIAL version 4.8 (http://prime.psc.riken.jp/) and annotated against the Human Metabolome Database (HMDB) version 5.0 (http://hmdb.ca) and the Kyoto Encyclopedia of Genes and Genomes (KEGG) Release 107.0 (https://www.kegg.jp). Differential metabolites, identified by a variable importance in projection (VIP) score greater than 1.0 from the orthogonal partial least squares discriminant analysis (OPLS-DA) model, a *p*-value less than 0.05 via a two-tailed *t*-test, and a fold change greater than 2 were subjected to pathway enrichment analysis using MetaboAnalyst 5.0 (https://metaboanalyst.ca).

### 2.7. Histopathological

Colon tissues were fixed in 4% paraformaldehyde (Sigma-Aldrich, USA, P6148), embedded in paraffin using a Leica HistoCore Pearl tissue processor (Leica Biosystems, Nussloch, Germany), and sectioned at a thickness of 5 μm with an RM2255 microtome (Leica, Wetzlar, Germany). Hematoxylin and eosin (H&E) staining was conducted using a KFBIO HES-1000 automated stainer (KFBIO, Yuyao, China, HES-1000). For immunofluorescence analysis, antigen retrieval was performed in citrate buffer (pH 6.0, 10 mM; Sigma-Aldrich, C1909), followed by blocking with 3% bovine serum albumin (BSA; Thermo Fisher, USA, 37525) and incubation with primary antibodies: anti-zonula occludens (ZO-1) (Invitrogen, Carlsbad, CA, USA, 40-2200, 1:1000) and anti-Occludin (Abcam, ab216327, 1:1000). Secondary antibodies, goat anti-rabbit IgG Alexa Fluor 488 (Thermo Fisher, A-11008, 1:500), were applied, and nuclei were counterstained with DAPI (Sigma-Aldrich, D9542). Images were captured using a Nikon Ts2-FL fluorescence microscope equipped with a 20× objective lens and NIS-Elements AR 5.30 software (Japan).

### 2.8. Fecal Microbiota Transplantation Validation Experiment

Frozen fecal samples from the CM-EVs group were resuspended in sterile PBS to create a 200 mg/2 mL bacterial suspension. This mixture was homogenized, filtered through 200, 400, and 800 mesh sieves, vortexed for 5 min, and centrifuged at 600× *g* for 5 min to remove insoluble materials. The supernatant was used for gavage. Twenty-four mice, kept at simulated 5500 m altitude, were divided into four groups and given a standard diet and water. The control group (*n* = 6) received 50 μL/kg/day of PBS, while the other groups received 25, 50, or 75 μL/kg/day of the bacterial suspension every two days for 15 days.

### 2.9. Statistical Analysis

Continuous data were expressed as mean ± standard deviation (SD; *n* = 6 biological replicates) and analyzed using GraphPad Prism 9.0 (GraphPad Software Inc., San Diego, CA, USA). Normality and homogeneity of variance were verified by Shapiro–Wilk test (α = 0.05) and Levene’s test, respectively. Intergroup differences were assessed via one-way ANOVA with Tukey’s post-hoc test, while metabolomic and microbiome datasets were analyzed using SIMCA-P 14.1 (Umetrics AB, Umeå, Sweden; v14.1.0.2045). Statistical significance was defined as adjusted *p* < 0.05.

## 3. Results

### 3.1. Extraction and Characterization of CM-EVs

CM-EVs were isolated and analyzed using standard methods. TEM showed spherical CM-EVs (Figure 2A), while NTA indicated sizes between 30 and 200 nm, peaking at 100 nm (Figure 2B). Western blot confirmed high levels of exosomal markers TSG101, CD63, and CD81 in CM-EVs, with minimal presence in whey (Figure 2C). SDS-PAGE with Coomassie Blue staining revealed a diverse protein profile, indicating CM-EVs’ functional complexity (Figure 2D).

### 3.2. Hypobaric Hypoxia-Induced Colonic Injury in AMS Mice and DAI Evaluation

Exposure to simulated high-altitude conditions (5500 m; 379 ± 2 mmHg) induced acute mountain sickness (AMS) in mice, as evidenced by a significant increase in the disease activity index (DAI) (*p* < 0.01) (Figure 3A,B). Hypoxic conditions resulted in notable colon damage, characterized by a reduction in colon length (*p* < 0.01), diminished mucosal thickness (*p* < 0.001), and decreased crypt depth (*p* < 0.01) (Figure 3C–G). Furthermore, immunofluorescence analysis demonstrated a significant decrease in the expression of tight junction proteins ZO-1 and Occludin (*p* < 0.5) (Figure 3H–J), suggesting that hypoxia contributes to the disruption of the intestinal barrier.

### 3.3. CM-EVs Attenuate Hypoxia-Induced Colonic Injury

Administering CM-EVs at a dosage of 0.5 mg/kg/day over a 15-day period significantly mitigated hypoxia-induced colon damage in mice. In comparison to PBS or camel milk, CM-EVs were found to reduce colon shortening and edema (*p* < 0.05, as illustrated in Figure 4A–D). Histological analysis revealed a preservation of mucosal structure, an increase in crypt depth (*p* < 0.01 compared to camel milk), and a reduction in inflammatory infiltration (refer to Figure 4E–I). Furthermore, CM-EVs restored the expression levels of ZO-1 and Occludin to near-normal, suggesting an enhancement in barrier function (*p* < 0.001 compared to hypoxia; see Figure 4J–O).

### 3.4. CM-EVs Modulate Gut Microbiota Composition

16S rRNA sequencing revealed distinct microbial changes across treatment groups. The CM-EVs group showed significantly higher α-diversity (Shannon index, *p* < 0.05 vs. CM group) and unique β-diversity patterns (Figure 5A–C). CM-EVs increased Firmicutes and decreased Bacteroidetes and Proteobacteria at the phylum level (*p* < 0.05 vs. PBS group). At the genus level, *Oscillospira* and *Ruminococcus* were elevated, with AF12 as a distinct taxon (Figure 5D). PICRUSt analysis indicated enriched pathways for carbohydrate metabolism, amino acid synthesis, and terpenoid biosynthesis (Figure 5E–G), suggesting CM-EVs may reprogram microbial metabolism.

### 3.5. CM-EVs Regulate Fecal Metabolite Profiles

An untargeted metabolomics analysis identified 617 metabolites exhibiting significant differences in abundance (*p* < 0.01). Principal Component Analysis (PCA) effectively differentiated between the PBS, CM, and CM-EVs groups (Figure 6A), with lipid-related molecules (32.36%) and organic acids (20.66%) constituting the primary components (Figure 6B,C). KEGG pathway analysis associated CM-EVs with bile secretion (*p* < 0.01), vitamin absorption (*p* < 0.05), and alpha-linolenic acid metabolism (*p* < 0.001) (Figure 6D–I). Pantothenic acid and naringenin demonstrated strong correlations with AF12 and *Oscillospira* (Spearman’s *r* > 0.6), suggesting potential microbiota–metabolite interactions contributing to the protective effects of CM-EVs. The correlation between gut microbiota and metabolites is illustrated in Figure 7.

### 3.6. CM-EVs Suppress Pro-Inflammatory Cytokines

In a simulated high-altitude environment (5500 m), fecal microbiota from the CM-EV group was transplanted into mice. The FMT improved colitis-like symptoms dose-dependently. Mice given moderate (50 μL/kg/day) and high doses (75 μL/kg/day) showed significantly less colon shortening compared to the PBS control group (*p* < 0.05) (Figure 8A,B). High doses also increased mucosal thickness (*p* < 0.05) (Figure 8C,D), suggesting that CM-EV microbiota effectively protected the gut from high-altitude stress.

## 4. Discussion

This study shows that camel milk-derived extracellular vesicles (CM-EVs) are highly effective in treating high-altitude hypoxia-induced colonic damage. CM-EVs are more effective than whole camel milk, significantly improving colonic mucosal barrier integrity, reducing disease severity (DAI, *p* < 0.01), and promoting a healthier gut microbiota (increased Lactobacillus and Bifidobacterium, decreased Enterobacteriaceae, *p* < 0.01). They not only alleviate symptoms but also restore gut homeostasis (Figure 9).

The successful isolation and characterization of CM-EVs highlight their potential as bioactive nanocarriers [27]. The spherical morphology and size distribution (30–200 nm) are consistent with typical exosome characteristics, and the strong expression of TSG101, CD63, and CD81 further confirms their exosomal identity [28,29,30,31]. The diverse protein profile, as revealed by SDS-PAGE, indicates that CM-EVs contain multifunctional cargo, which may contribute to their therapeutic effects [32,33,34]. In contrast to camel milk, which encounters challenges related to thermal stability and bioavailability, CM-EVs are able to bypass digestive degradation and deliver intact biomolecules to intestinal cells via endocytosis, thereby enhancing their functional efficacy [35,36].

Exposure to hypobaric hypoxia, simulating an altitude of 5500 m, resulted in significant acute colonic damage in mice, characterized by mucosal atrophy, crypt degeneration, and compromised tight junction integrity. Pre-treatment with extracellular vesicles derived from CM-EVs markedly mitigated these pathological changes, effectively restoring mucosal architecture and the expression of tight junction proteins. In addition to structural repair, hypoxia induced a distinct alteration in gut microbiota, as evidenced by a decreased Firmicutes/Bacteroidetes ratio [37,38]. The administration of CM-EVs successfully counteracted this dysbiotic shift, restoring microbial alpha diversity and enriching beneficial genera such as *Oscillospira* and *Ruminococcus*, which are known for their butyrate production and anti-inflammatory properties [39,40,41,42]. The exclusive enrichment of the AF12 genus in the CM-EV-treated group suggests a novel role for this genus in fatty acid metabolism under hypoxic conditions, warranting further investigation. Functional prediction analysis using PICRUSt indicates that CM-EVs reprogram the gut microbial metabolic potential, significantly enhancing pathways related to carbohydrate metabolism and terpenoid biosynthesis [43,44,45]. These modifications may contribute to enhanced host resilience. Our integrated pathway modulation analysis demonstrates that CM-EVs exert protective effects primarily via the FXR/NF-κB pathway, which restores gut barrier integrity and remodels microbial communities. Additionally, CM-EV treatment improves host metabolic profiles, as evidenced by enhanced bile acid homeostasis and optimized amino acid metabolism.

Untargeted metabolomics further substantiates the significant influence of CM-EVs on fecal metabolites, with a particular emphasis on lipid-related molecules and organic acids. The enrichment of pathways associated with bile secretion and alpha-linolenic acid metabolism supports previous research linking these pathways to the repair of the intestinal barrier [46,47,48,49,50]. Additionally, the synergistic microbiota–metabolite axis is underscored by the robust correlations identified between key metabolites, such as naringenin and pantothenic acid, and the enriched beneficial microbes, AF12 and *Oscillospira*. This study identifies microbiota-derived metabolites as key contributors to the anti-inflammatory effects of CM-EVs. Specifically, pantothenic acid has been demonstrated to suppress Th17 cell differentiation by inhibiting the PKM2-STAT3 signaling pathway, while naringenin has been observed to mitigate inflammation mediated by the NF-κB pathway [50,51]. Consequently, CM-EVs facilitate a synergistic interaction between the microbiota and metabolites, thereby alleviating colitis induced by hypoxic conditions. CM-EVs exhibit enhanced efficacy over whole camel milk in mitigating colonic damage induced by hypobaric hypoxia. This protective effect is mediated through multifaceted mechanisms, including the restoration of the intestinal barrier, modulation of the gut microbiota, and metabolic reprogramming. Consequently, CM-EVs represent promising novel nutraceuticals for high-altitude adaptation, exemplifying the potential of nanotechnology to augment the bioactivity of components sourced from traditional foods.

The therapeutic significance of microbiota remodeling was further substantiated through FMT. The successful transfer of microbiota modulated by CM-EVs to recipient mice replicated colonoprotective effects, thereby demonstrating the causal involvement of microbial communities. This interplay between vesicle-mediated molecular signaling and microbial dynamics underscores a previously unrecognized axis for protection against hypoxia. Human milk-derived extracellular vesicles (HM-EVs) play a crucial role in regulating intestinal function. These vesicles are internalized by intestinal epithelial cells in the neonatal gut, predominantly via clathrin-mediated endocytosis, which facilitates intestinal maturation, enhances barrier integrity, and reduces inflammation [52,53]. Empirical studies suggest that a deficiency in extracellular vesicles derived from cow milk leads to impaired intestinal barrier function, as evidenced by increased mucosal permeability, decreased villus height, and reduced crypt depth. These deficiencies can result in intestinal health issues and hinder weight gain in neonates [53,54]. Furthermore, microRNAs encapsulated within these vesicles, such as miRNA-148a, contribute to the inhibition of the NF-κB signaling pathway, thereby lowering the risk of necrotizing enterocolitis (NEC) and supporting intestinal development and overall immune function [55,56,57].

Despite advancements, several limitations remain: Reliance on murine models limits human relevance; unidentified bioactive components in CM-EVs need analysis; long-term effects on microbial stability and mucosal integrity are unclear; and human biochemical response variability is unassessed. Researchers extracted extracellular vesicles from first-time birthing Qaidam Bactrian camels, and future studies should control for camel breed and parity effects on EV content and improve bioinformatics analysis. CM-EVs hold promise for gastrointestinal health due to nanoscale delivery and camel milk’s anti-inflammatory properties. Key applications include the following: (1) anti-inflammatory agents for IBD and mucosal repair; (2) strengthening intestinal barriers for “leaky gut syndrome”; (3) balancing gut microbiota for IBS; and (4) targeted oral treatments. CM-EVs from natural sources show promise as new treatments for gastrointestinal issues and as functional food ingredients. While challenges in scalable production, standardization, and clinical validation exist, technological advances and cross-disciplinary collaboration are expected to support their industrialization for therapeutic and nutritional products.

## 5. Conclusions

In summary, CM-EVs provide a multi-faceted approach to address hypoxia-induced colonic injury by repairing mucosal damage, correcting dysbiosis, and restoring metabolic balance through FXR/NF-κB pathway modulation. Although the murine model offers strong mechanistic insights, future studies should aim to validate these findings in human-relevant models, thoroughly profile the bioactive components of CM-EVs, and refine production and delivery methods for practical nutraceutical use. This research highlights nanotechnology’s potential to boost the therapeutic benefits of traditional foods.

## Figures and Tables

**Figure 1 nutrients-17-02431-f001:**
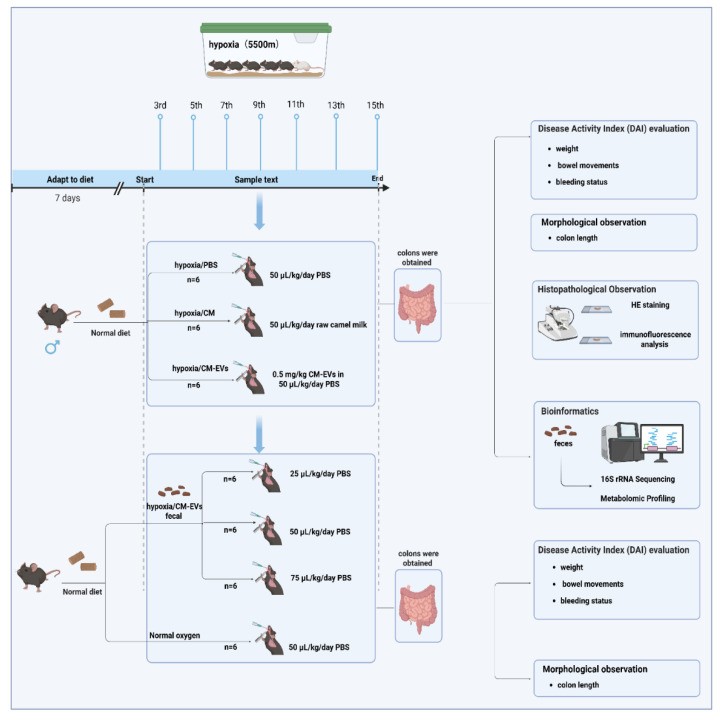
Specific experimental design flowchart.

**Figure 2 nutrients-17-02431-f002:**
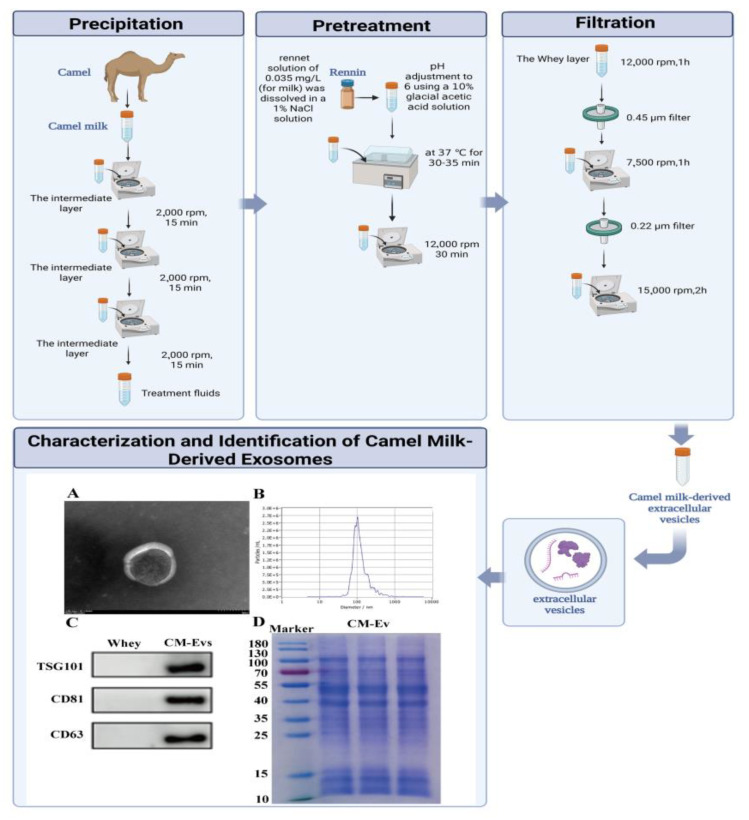
CM-EVs Analysis. (**A**) Transmission electron microscopy revealed CM-EVs morphology; scale bar: 100 nm. (**B**) NTA results showed particles mainly 40–100 nm in diameter. (**C**) Western blot identified exosome surface proteins CD81, CD63, and TSG101. (**D**) SDS PAGE displayed band profiles of CM-EVs proteins.

**Figure 3 nutrients-17-02431-f003:**
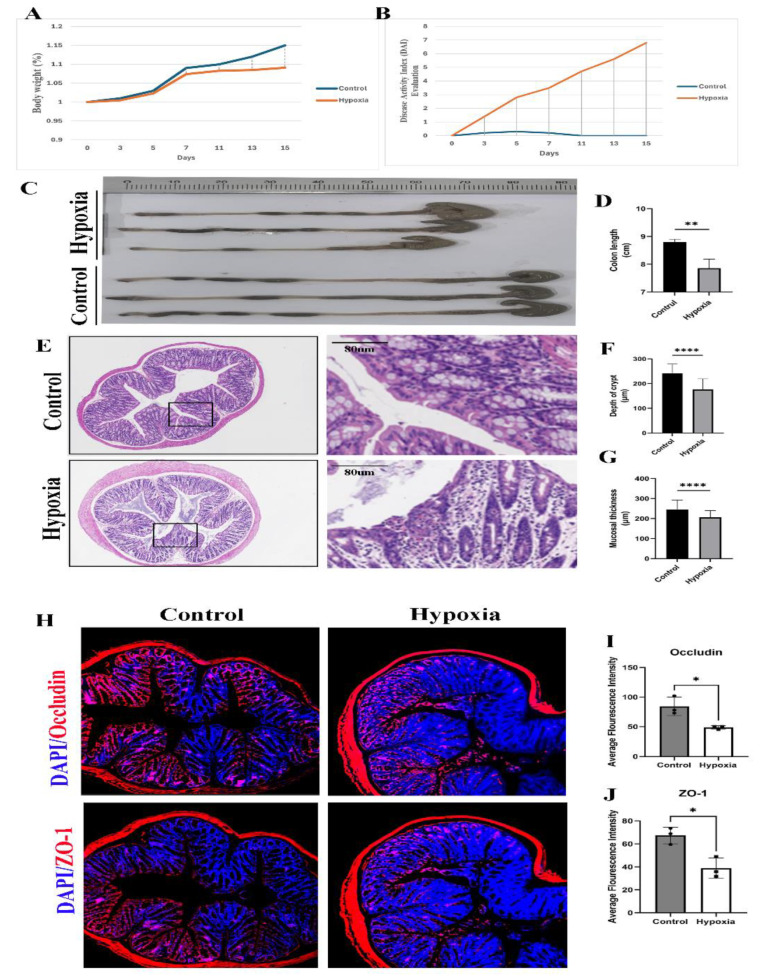
Effects of acute hypobaric hypoxia on colon health. (**A**) Mouse body weight changes. (**B**) Disease index variations. (**C**) Colon length alterations. (**D**) Colon length data analysis. (**E**) Colonic histopathology via H&E staining; scale 400 µm/80 µm. (**F**) Colonic mucosa thickness. (**G**) Colonic crypt depth. (**H**) Immunofluorescence of ZO-1 and Occludin for colonic barrier function; green indicates ZO-1 and Occludin, and blue indicates DAPI; scale bar 200 µm. (I) ZO-1 expression level statistics. (**J**) Occludin expression level statistics. Note: *, *p* < 0.5; **, *p* < 0.01; ****, *p* < 0.0001.

**Figure 4 nutrients-17-02431-f004:**
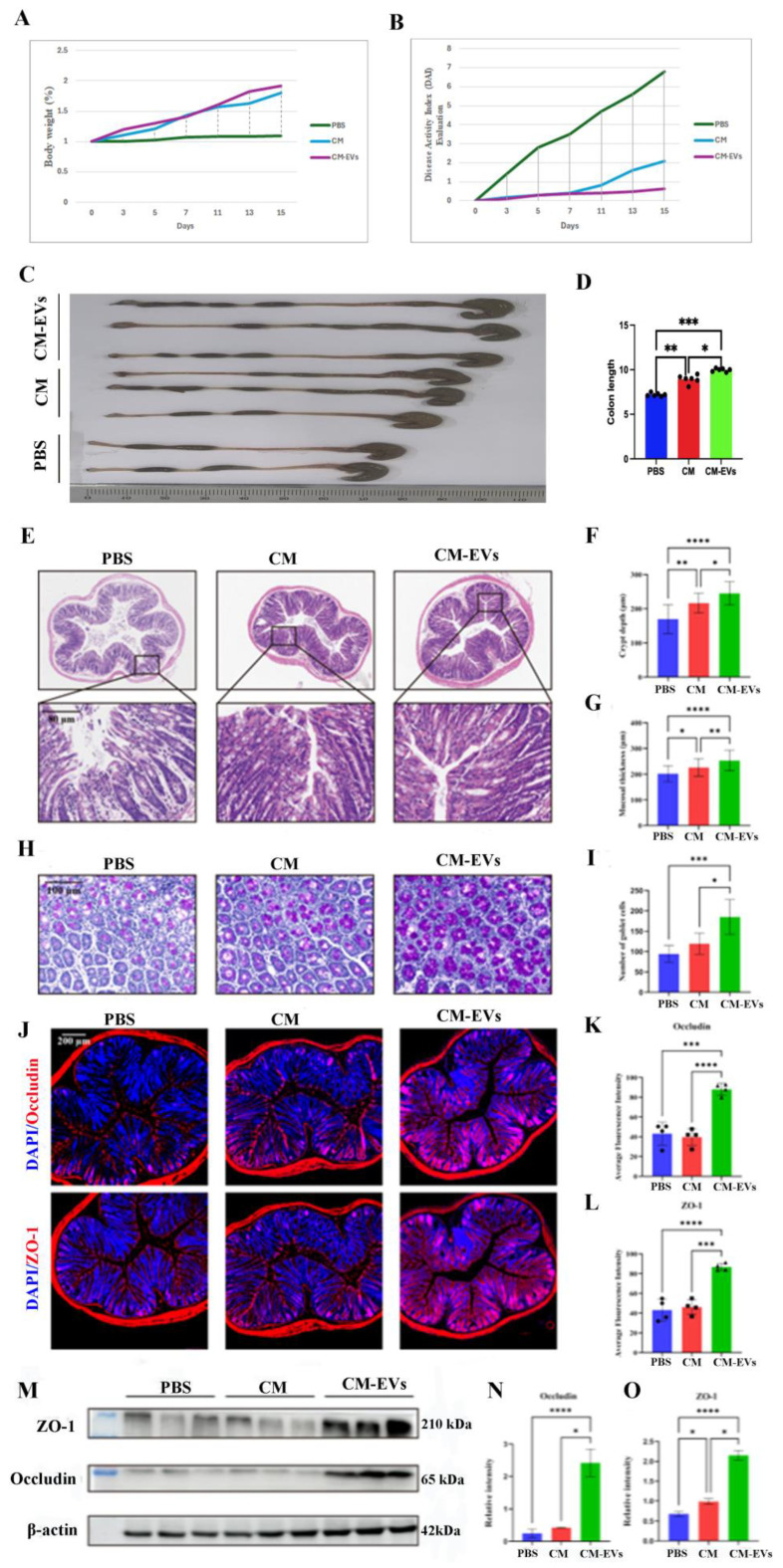
Experiment on mice under low oxygen with drug administration. (**A**) Mouse weight changes. (**B**) Disease index changes. (**C**) Colon length changes. (**D**) Colon length data analysis. (**E**) H&E staining of colon tissue, scale bar: 80 µm. (**F**) Colonic mucosa thickness. (**G**) Colonic crypt depth. (**H**) PAS-stained images. (**I**) Goblet cell density. (**J**) ZO-1 and Occludin immunofluorescence on colonic barrier; green: ZO-1/Occludin, blue: DAPI; scale bar: 200 µm. (**K**) ZO-1 expression statistics. (**L**) Occludin expression statistics. (**M**) Western blot of ZO-1 and Occludin. (**N**) ZO-1 expression via Western blot. (**O**) Occludin expression via Western blot. Note: *, *p* < 0.5; **, *p* < 0.01; ***, *p* < 0.001; ****, *p* < 0.0001.

**Figure 5 nutrients-17-02431-f005:**
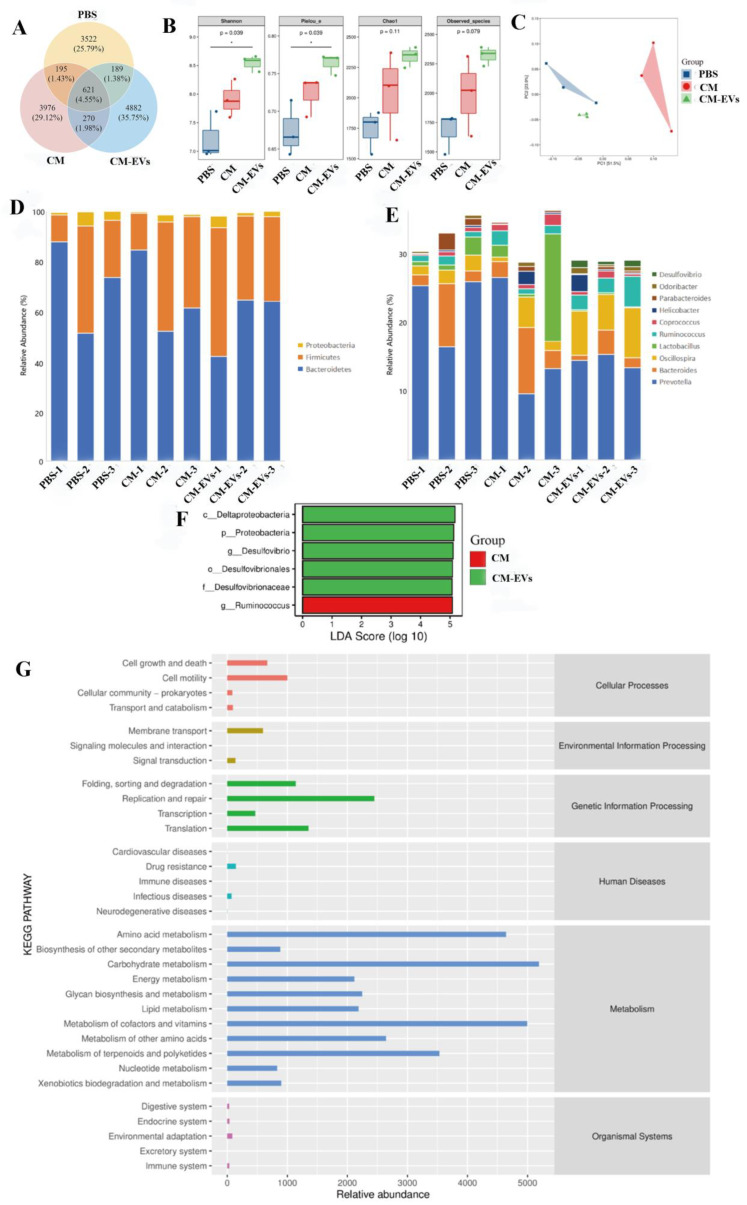
Gut Microbiota Structure Comparison. (**A**) Venn diagram of unique/shared bacterial genera. (**B**) Changes in fecal microbial α-diversity post-intervention. (**C**) PCA of UniFrac distances showing group differences in gut microbiota abundance. (**D**) Top 10 phyla relative abundance. (**E**) Top 10 genera relative abundance. (**F**) LDA score > 3 highlights group-specific abundance. (**G**) Predicted 16S rRNA metabolic pathways in KEGG via PICRUSt2. Note: *, *p* < 0.5.

**Figure 6 nutrients-17-02431-f006:**
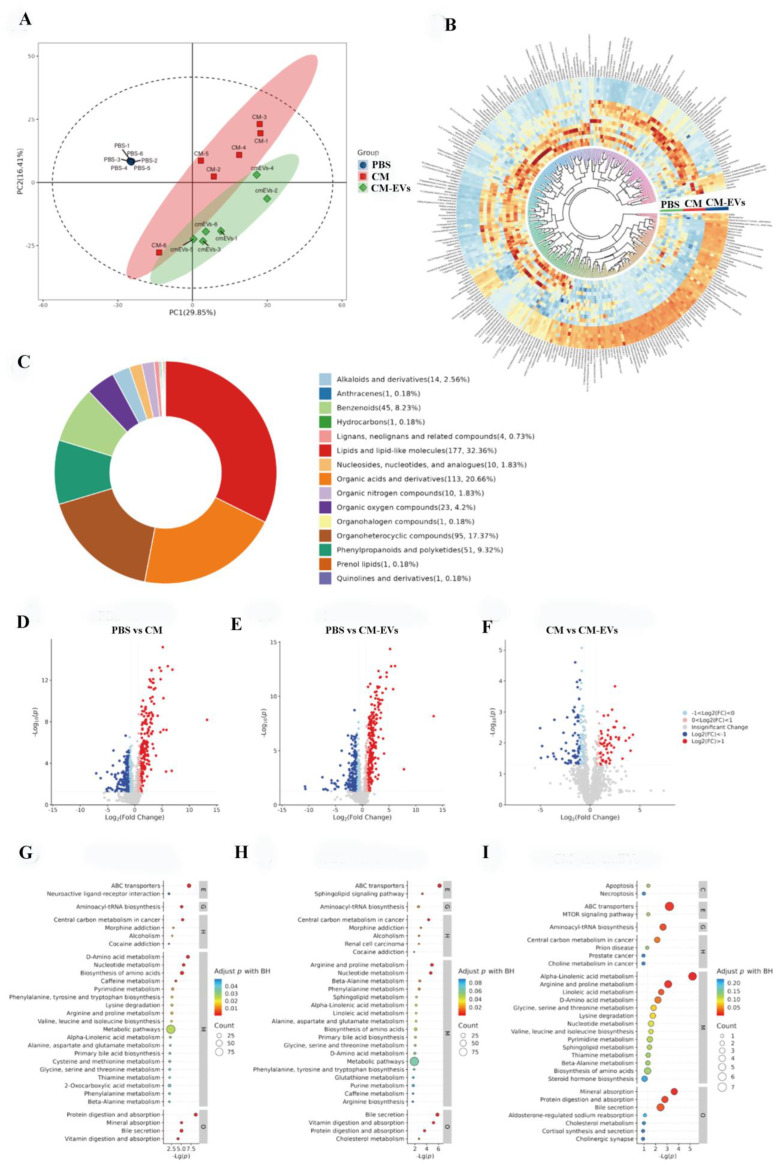
Fecal Metabolites Analysis. (**A**) PCA shows group differences. (**B**) Heat maps of top 500 metabolites across groups. (**C**) Metabolite classification diagram. (**D**–**F**) Volcano plots comparing differential metabolites between PBS vs. CM, PBS vs. mEVs, and CM vs. mEVs. (**G**–**I**) KEGG pathway enrichment for PBS vs. CM, PBS vs. mEVs, and CM vs. mEVs.

**Figure 7 nutrients-17-02431-f007:**
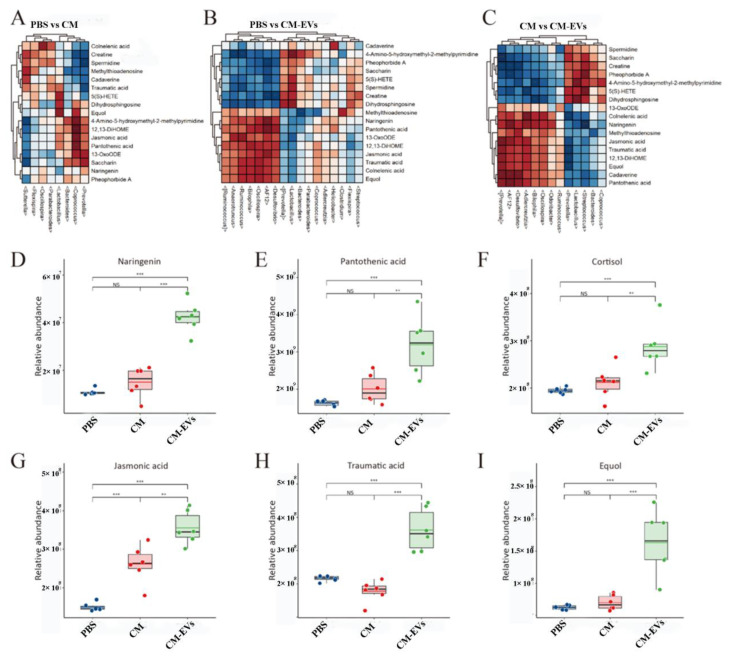
Gut microbiota and metabolite correlation. (**A**–**C**) Heat map. (**D**–**I**) Histogram of metabolites’ relative abundance linked to gut microbiota. Note: **, *p* < 0.01; ***, *p* < 0.001.

**Figure 8 nutrients-17-02431-f008:**
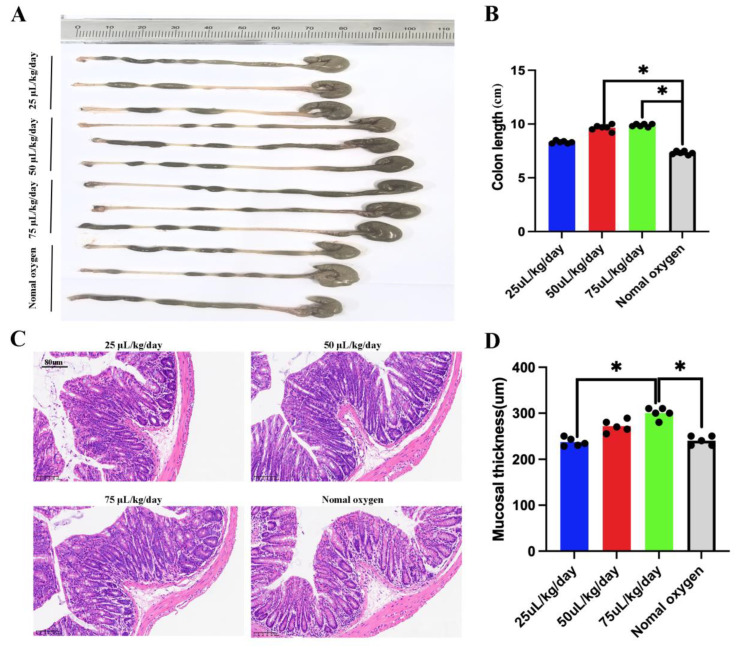
The FMT improved colitis-like symptoms dose-dependently. (**A**) Colon length alterations. (**B**) Colon length data analysis. (**C**) Colonic histopathology via H&E staining, scale bar: 80 µm. (**D**) Colonic mucosa thickness. Note: *, *p* < 0.5.

**Figure 9 nutrients-17-02431-f009:**
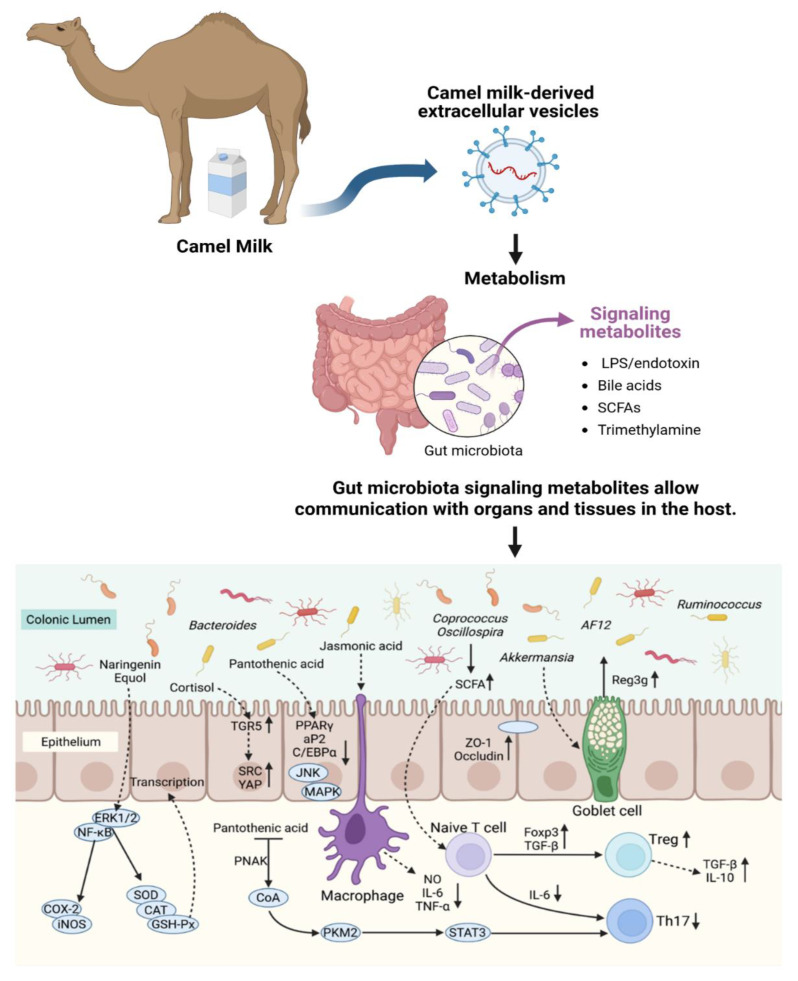
Schematic diagram of gut microbial structure changes involved in colonic protection.

## Data Availability

The original contributions presented in this study are included in this article/Appendix A. Further inquiries can be directed to the corresponding author.

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
