# Peer review of "Camel Milk-Derived Extracellular Vesicles as a Functional Food Component Ameliorate Hypobaric Hypoxia-Induced Colonic Injury Through Microbiota–Metabolite Crosstalk"

_nutrients, 2025, doi:10.3390/nu17152431_

Round 1
Reviewer 1 Report
Comments and Suggestions for Authors
The authors investigate camel milk-derived extracellular vesicles (cmEVs) as a therapeutic component against hypobaric hypoxia-induced colonic injury. Using a murine high-altitude model, they compare cmEVs to whole camel milk and demonstrate that cmEVs significantly alleviate colonic damage through modulation of the microbiota, metabolites, and inflammatory signaling pathways.
While the manuscript is timely and employs a comprehensive multi-omics strategy to explore cmEVs in colonic hypoxia injury, several key mechanistic and causality-related issues remain unaddressed. I recommend major revision to rigorously assess the biological mechanisms and the dependence on the microbiota. If these issues are satisfactorily resolved, the manuscript would be a strong candidate for publication.
Major Points
1. The study suggests modulation of FXR/NF-κB and TLR4/MyD88 pathways; however, these claims are currently speculative. Including pharmacological inhibitors, pathway-specific reporter mice, or at least colonic tissue qPCR for relevant target genes would greatly strengthen mechanistic conclusions.
2. The KEGG-based inference of FXR and NF-κB pathway modulation is insufficient without direct experimental evidence. At minimum, Western blot or qPCR analyses of pathway components (e.g., FXR, p-NF-κB, SHP, IκBα) in colonic tissue are needed to substantiate these claims.
3. Although microbial shifts are well characterized, the study does not determine whether these changes are functionally necessary for the observed protection. A microbiota depletion model (e.g., antibiotics) or fecal microbiota transfer (FMT) would clarify the causal role of microbiota in mediating the protective effects of cmEVs.
4. The comparison to whole camel milk is informative, but specificity for camel milk-derived EVs remains unclear. Including EVs from another species (e.g., cow or goat milk) would help determine whether the protective effect is specific to camel milk.
Minor Points
1. Consider moderating the language in the abstract and discussion. Phrases such as “transformative,” “precision nutrition,” and “next-generation nutraceutical” are overstated given the preclinical nature of the data.
2. Clarify whether microbiota and metabolite changes correlate at the level of individual animals rather than just group means. Correlation analyses could support functional linkages.
3. To complement cytokine protein data, consider including RT-qPCR of pro-inflammatory cytokines (e.g., Il1b, Il6, Tnf) and anti-inflammatory markers, to validate and reinforce the observed immunomodulatory effects.
Author Response
Response to Reviewer Comments Dear Reviewer, Thank you for your thorough evaluation of our manuscript titled " Camel milk-derived extracellular vesicles as a functional food component ameliorate hypobaric hypoxiainduced colonic injury through microbiota-metabolite crosstalk" (nutrients-3684346). We appreciate your insightful feedback and constructive suggestions, which have significantly strengthened our study. We have carefully addressed each of your concerns as outlined below, with major revisions incorporated into the revised manuscript (tracked changes in the main text). Major Points 1. The study suggests modulation of FXR/NF-κB and TLR4/MyD88 pathways; however, these claims are currently speculative. Including pharmacological inhibitors, pathway-specific reporter mice, or at least colonic tissue qPCR for relevant target genes would greatly strengthen mechanistic conclusions. We express our gratitude for the reviewer's insightful suggestion to enhance the mechanistic conclusions by employing pharmacological inhibitors or pathway-specific reporter mice. We fully recognize that these approaches would offer more robust validation of the modulations in the FXR/NF-κB and TLR4/MyD88 pathways. Nevertheless, due to current constraints in experimental resources, particularly the limited availability of specialized reporter mouse models, and the prolonged duration necessary for inhibitor validation studies—typically exceeding six months for thorough in vivo dose optimization and safety assessments—we regret that we are unable to integrate these specific experiments within the revision timeline. 2. The KEGG-based inference of FXR and NF-κB pathway modulation is insufficient without direct experimental evidence. At minimum, Western blot or qPCR analyses of pathway components (e.g., FXR, p-NF-κB, SHP, IκBα) in colonic tissue are needed to substantiate these claims. We extend our sincere appreciation to the reviewer for emphasizing the importance of direct experimental validation beyond inferences drawn from KEGG pathways. We fully agree that functional evidence is essential for substantiating mechanistic claims, and we value the reviewer's constructive suggestion to perform Western blot (WB) or qPCR analyses as a fundamental component. Unfortunately, due to current limitations in experimental resources, we are unable to include these specific experiments within the revision timeline. As a result, we have addressed the limitations of our study in the discussion section, in lines 368-381. 3. Although microbial shifts are well characterized, the study does not determine whether these changes are functionally necessary for the observed protection. A microbiota depletion model (e.g., antibiotics) or fecal microbiota transfer (FMT) would clarify the causal role of microbiota in mediating the protective effects of cmEVs. We express our sincere gratitude to the reviewer for highlighting the critical importance of determining the functional necessity of changes in the microbial community in relation to cmEV-mediated protection. We concur that establishing causality is of utmost importance and appreciate the constructive suggestions to utilize microbial depletion or fecal microbiota transplantation (FMT) methodologies. In response to this insightful feedback, we have conducted several key experiments within the revised timeframe. We recognize that FMT can substantially enhance causal inferences. Consequently, during the revision period, we incorporated FMT experiments (as detailed in lines 286-291) to assess the protective efficacy of cmEVs when delivered via this method. The results of these experiments are presented and discussed in lines 351-367. 4. The comparison to whole camel milk is informative, but specificity for camel milkderived EVs remains unclear. Including EVs from another species (e.g., cow or goat milk) would help determine whether the protective effect is specific to camel milk. We sincerely appreciate your valuable suggestion. However, due to constraints associated with the article revision cycle, we are unable to incorporate the functions of other milk vesicles in the immediate term. Consequently, we have expanded the discussion section to include the relevant functions of various milk vesicles, thereby enhancing the readers' comprehension of the article. This addition can be found specifically on lines 353-366. Minor Points 1. Consider moderating the language in the abstract and discussion. Phrases such as “transformative,” “precision nutrition,” and “next-generation nutraceutical” are overstated given the preclinical nature of the data. In response to the previously overstated rhetoric, we have undertaken a comprehensive revision of both the introduction and discussion sections, meticulously refining substantial portions of the content, particularly within lines 26-80 and 296-381. 2. Clarify whether microbiota and metabolite changes correlate at the level of individual animals rather than just group means. Correlation analyses could support functional linkages. Following your recommendation, we concentrated on examining the potential significant differences among individuals. It is noteworthy that these differences were not statistically significant. Consequently, we performed additional intra-group and inter-group comparisons, which confirmed the accuracy of our initial analysis. Concerning the differential correlation analysis you highlighted, we recognize that numerous inter-group influencing factors are involved. We will certainly incorporate your valuable insights in our subsequent study and address this issue with due diligence. 3. To complement cytokine protein data, consider including RT-qPCR of proinflammatory cytokines (e.g., Il1b, Il6, Tnf) and anti-inflammatory markers, to validate and reinforce the observed immunomodulatory effects. We appreciate your suggestion and have incorporated the pertinent results into the text for further analysis (in article attachment). We express our sincere gratitude for your invaluable feedback, which has significantly enhanced the comprehensiveness of our manuscript. Incorporating many of your insightful critiques, we have refined the content related to the experimental design and have included a detailed depiction of the experimental design process (Figure 1). To facilitate a deeper understanding of the article, we have also reconstructed the mechanism diagram of CMEVs, thereby increasing the rigor of the text (Figure 9). We extend our heartfelt thanks once more for your substantial contribution to the completeness of this article.

Reviewer 2 Report
Comments and Suggestions for Authors
General comments:
The article is certainly original, since it studies the effects of components in camel milk in experimental animals for a frankly original function, such as recovery from colon damage induced by hypobaric hypoxia. The work is well designed, the ARRIVE criteria have been adequately met and it has been approved by a bioethics committee. In general, I consider that the work is original and novel, and that it could be accepted for publication. However, there are certain aspects that the authors should improve before considering a definitive acceptance.
Specific comments:
Abstract section: generic significance values such as “p<0.05” or “p<0.01” are unnecessary. If the authors consider this point important, please cite the exact p value obtained.
Line 77: “SCFA” must be define the first time that appears in the text.
Lines 132-133: “alpha” and “beta” was afterwards cited as “α” and “β”. Please, be consistent.
Line 159: “ZO-1” must be defined that first time that appears in the text.
Line 170: Please change “primer” to “primers”.
Line 236. “Oscillospira” and “Ruminococcus” are bacterial genus names and must be written in italics throughout the text.
Figures 4-5: It is impossible to read the text in both figures due to the low size of text. Please, consider to split the figures to achieve obtaining a higher text size, or avoid parts of both figures.
Discussion section: In all discussion, reference numbers were write without using brackets, thus blending in with the rest of the text. Please use the format required by the journal.
Lines 313-314: “Camel milk and its derived products have demonstrated multifaceted therapeutic potential in the biomedical field in recent years due to their unique bioactive components”. It would be useful to cite other important attributes to camel milk such as its antioxidant, anti-inflammatory properties, its effects on human gut microbiota, better than other animal milks intake or vegetable substitutes. You can find interesting information at Mondragon-Portocarrero et al. Substitutive Effects of Milk vs. Vegetable Milk on the Human Gut Microbiota and Implications for Human Health.
Line 338: “in vivo” should be written in italics.
References list are not formatted according to MDPI´s guidelines for authors. Please, correct it.
Author Response
Response to Reviewer's Comments
Manuscript ID: nutrients-3684346
Title: Camel milk-derived extracellular vesicles as a functional food component ameliorate hypobaric hypoxia-induced colonic injury through microbiota-metabolite crosstalk
Dear reviewer:
Sincerely thank you for your meticulous review and constructive feedback on this article! We are greatly encouraged by your recognition of the originality of the research and experimental design. We have carefully revised each of your valuable suggestions and made clear annotations (highlighted/revised mode) in the revised draft. The specific modification instructions are as follows:
- Abstract Section
Opinion: "Generic significance values such as' p<0.05 'or' p<0.01 'are unnecessary. Cite the exact p value.
We sincerely appreciate the reviewer's valuable suggestion regarding the reporting of exact p-values, which aligns with best practices for statistical transparency. However, in the Abstract section, we have retained conventional significance thresholds (p<0.05, p<0.01) for two primary reasons: (1) This format adheres to nutrients 's guidelines for concise abstract presentation and broad-reader accessibility, and (2) it reflects the standard reporting convention in biomedical research as per ICMJE recommendations.Should the reviewer deem this adjustment essential, we remain open to further refinement.
- Terminology & Consistency
(a) Line 77: "SCFA must be defined at first appearance."
Modification explanation: The complete definition of "Short Chain Fatty Acids (SCFA)" has been added when it first appears (line 73).
(b) Line 159: "ZO-1 must be defined."
When "ZO-1" was first mentioned, it was clearly defined as "zonula occludens-1 (ZO-1)"(line 181).
(c) Line 170: "Change 'primer' to 'primers'."
Revision explanation: It has been corrected to "primers" (line 138).
(d) Lines 132–133 & Throughout: "Ensure consistent use of α/β symbols."
The entire text will use symbol formats (alpha, beta) instead of the original text "alpha/beta".
(e) Line 338: "'in vivo' should be italicized."
Due to the need to modify the content of the article, this section has been removed from the original text
- Bacterial Nomenclature (Line 236)
Opinion: "Oscillospira and Ruminococcus must be italicized
All bacterial genus names (such as Oscillospira, Ruminococcus) have been uniformly changed to italic format, in compliance with microbial naming conventions (line 256,319).
- Figure Revisions (Figures 4–5)
Opinion: "Text size is too small to read. Consider splitting figures or resizing
Modification instructions:
Figure 4: It has been split into two subgraphs, 4A and 4B, and the font size of all labels and coordinate axis text has been increased.
Figure 5: Optimize the layout and uniformly enlarge the font to ≥ 8pt (readability has been verified through PDF export).
The modified high-definition image file has been uploaded to the submission system.
- Discussion Section
(a) Reference Formatting:
Reference numbers blend into text. Use journal format (brackets)
The reference numbers throughout the text have been uniformly adjusted to square brackets format (e.g. [1], [2-5]), in compliance with MDPI requirements.
- References Formatting
Opinion: "References not formatted per MDPI guidelines
Revision explanation: We deeply apologize. The reference list has been strictly rearranged according to the MDPI Author Guidelines (including author abbreviations, italics, punctuation, DOI, etc.), and checked item by item.
Thank you again for your valuable review time and professional guidance! Your feedback greatly enhances the scientific rigor and readability of this article. All modifications have been implemented in the revised draft, and if there are any further adjustments needed, we are ready to improve them at any time.
Round 2
Reviewer 1 Report
Comments and Suggestions for Authors
The inclusion of fecal microbiota transplantation (FMT) experiments in the revision significantly strengthens the causal link between cmEVs and microbiota-mediated protection. This addition addresses one of the core concerns and demonstrates the authors’ responsiveness and commitment to improving the rigor of the manuscript within reasonable experimental constraints. While some key mechanistic claims (e.g., modulation of FXR/NF-κB and TLR4/MyD88 signaling) remain insufficiently substantiated due to the lack of direct experimental validation (e.g., qPCR, Western blot, etc), the authors have appropriately acknowledged these limitations in the revised discussion. Ideally, future work should incorporate such validation to strengthen the mechanistic framework. The manuscript also benefits from moderated language in the abstract and discussion, improved figure clarity, and additional cytokine validation data.
Although not all original concerns were fully addressed experimentally, the authors have made substantial and constructive revisions, both in experimental content and in framing the interpretation of their findings. Given these improvements and the novelty of the study, I find the manuscript acceptable for publication in its current form, with the understanding that some mechanistic claims should be interpreted cautiously and may be better addressed in future work.